# Correlations Between Amelioration of Rotenone-Induced Parkinson’s Symptoms by *Amomum tsaoko* Flavonoids and Gut Microbiota in Mice

**DOI:** 10.3390/ijms26041676

**Published:** 2025-02-16

**Authors:** Li Liu, Yan Zhao, Weixing Yang, Lixiang Han, Xiaohan Mo, Jun Sheng, Yang Tian, Xiaoyu Gao

**Affiliations:** 1Yunnan Key Laboratory of Precision Nutrition and Personalized Food Manufacturing, Yunnan Agricultural University, Kunming 650201, China; 2022210085@stu.ynau.edu.cn (L.L.); shengj@ynau.edu.cn (J.S.); 2College of Food Science and Technology, Yunnan Agricultural University, Kunming 650201, China; 2021110006@stu.ynau.edu.cn (W.Y.); 15368025745@163.com (L.H.); 15187349230@163.com (X.M.); 3Engineering Research Center of Development and Utilization of Food and Drug Homologous Resources, Ministry of Education, Yunnan Agricultural University, Kunming 650201, China; 4Division of Science and Technology, Yunnan Agricultural University, Kunming 650201, China; 2021013@ynau.edu.cn

**Keywords:** movement disorders, gastrointestinal dysfunction, inflammation, gut barrier, constipation, *Desulfovibrio*, *Provotellaceae*, *Lachnospiraceae*, *Bifidobacterium*, *Faecalibaculum*

## Abstract

Parkinson’s disease (PD) is the second most common neurodegenerative disease, but the existing therapeutic drugs for PD have limitations; thus, there is an urgent need to discover new methods of prevention and treatment. *Amomum tsaoko* Crevost et Lemarie (AT) is a classic traditional Chinese medicine and food. Its main pharmacological effect is the regulation of the gastrointestinal tract. To date, no studies on the use of AT or its extracts to treat PD have been reported. In this study, a rotenone-induced PD mouse model was utilized to evaluate the protective effect of *Amomum tsaoko* flavonoids (ATFs) and to elucidate the role of the gut microbiota in this effect. The results demonstrated that ATFs not only ameliorated the motor and constipation symptoms but also reduced the loss of nigrostriatal dopaminergic neurons. Furthermore, ATFs reduced the expression of inflammation-related genes (*TNF-α*, *IL-1β*, *IL-6*, *COX-2*, and *MCP-1*) and increased the expression of gut barrier-related genes (*Muc-2*, *ZO-1*, *Occludin*, *Claudin3*, and *Claudin4*) in the colon. Notably, ATFs were able to reverse rotenone-induced gut dysbiosis, including a significant decrease in the abundance of conditionally pathogenic bacteria (*Desulfovibrio*, *Provotellaceae UCG-001*, the *Lachnospiraceae*_*NK4A136_group*, *norank_f_Erysipelotrichacea*, and the *Eubacterium nodatum group*) and an increase in the abundance of probiotics (*Bifidobacterium* and *Faecalibaculum*). Interestingly, these genera were found to be significantly associated with PD motor symptoms and constipation indicators. This suggests that ATFs have the potential to alleviate PD symptoms through the modulation of gut microbes. These findings provide a solid foundation for further investigations into the anti-PD mechanism of ATFs and their potential in the prevention and treatment of PD.

## 1. Introduction

Parkinson’s disease (PD) is the second most common neurodegenerative disease. It is estimated that more than 10 million people worldwide suffer from PD [1]. PD patients exhibit typical motor symptoms, including slow movements, rigidity, resting tremors, and postural instability [2]. In addition, a variety of non-motor symptoms have been reported, including gastrointestinal (GI) dysfunction such as constipation, dysphagia, nausea, and vomiting. These non-motor symptoms have a more pronounced negative impact on the patients’ quality of life [3]. The principal pathological features of PD are the selective loss of dopaminergic (DA) neurons in the substantia nigra (SN) of the midbrain and the aggregation of α-synuclein (α-syn) in the remaining neurons to form Lewy bodies [4]. There is increasing evidence that the abnormal aggregation of α-syn, mitochondrial dysfunction, oxidative stress, abnormal regulation of apoptosis, and neuroinflammation contribute to the disease process [5].

Presently, the primary treatment for PD is drug-based, with levodopa preparations (e.g., Madopar) being the most commonly prescribed drug to alleviate the symptoms of this condition. These preparations are capable of supplementing the deficiency of DA in the brain. Nevertheless, such pharmaceuticals have been demonstrated to be ineffective in halting or reversing the loss of DA neurons, and they do not prevent the progression of the disease [6]. Consequently, searching for medications that exhibit both definitive efficacy and minimal adverse effects has emerged as a pivotal area of the current research.

Recently, there have been advancements in the research related to flavonoids in the treatment of PD. Flavonoids are a class of natural compounds that are widely found in plants and exhibit various biological activities, including antioxidant, anti-inflammatory, and antiviral properties [7,8]. In addition, flavonoids have been demonstrated to protect mitochondrial function [9], inhibit neuronal apoptosis [10], and modulate neuroinflammatory responses [11]. Studies have reported that flavonoids possess the capacity to modulate the composition and function of the gut microbiota [12], impacting intestinal barrier function [13], immune regulation [14], and metabolite production [15], which are known to be involved in the development and progression of PD. For example, some flavonoids (catechins and quercetin) can increase the number of beneficial bacteria, inhibit the growth of harmful bacteria, and reduce the production and absorption of harmful substances, thereby reducing the adverse effects on the nervous system [16].

*Amomum tsaoko* Crevost et Lemarie (AT) has a long history of application in traditional Chinese medicine and as a spice [17]. AT is rich in flavonoids, and studies have indicated that *Amoma tsaoko* flavonoids (ATFs) possess antioxidant and anti-inflammatory properties. In an in vitro cellular assay, ATFs were able to reduce intracellular indicators of oxidative stress and attenuate the cellular damage [18]. In cell models of inflammation, ATFs have been shown to inhibit the release of inflammatory factors, thereby exerting anti-inflammatory effects [19]. In addition, Hu et al. [20] discovered that ATFs were able to alleviate loperamide-induced constipation in mice by modulating the gut microbiota and its related metabolites. However, whether ATFs have the potential to exert neuroprotective effects and become a candidate drug for relieving PD has not been studied. Therefore, in the present study, we investigated the anti-PD effects of ATFs and their effects on rotenone-induced intestinal inflammation, intestinal barrier damage, and intestinal microbiota dysregulation in PD mice. This study is expected to provide a theoretical basis for the development and application of functional products containing ATFs for neuroprotective therapy.

## 2. Results

### 2.1. Effect of ATF Treatment on Body Weight of PD Mice

As illustrated in Figure 1B–D, the body weight of the mice in each group exhibited an upward trend during the course of the experiment. A notable observation was the significantly higher body weight of the mice in the ATF group compared to rotenone-induced PD mice after 45 days (Figure 1B, *p* < 0.05). At the same time, the average water intake and food intake of each group were also within the normal fluctuation range, and there was no significant difference between the groups (Figure 1C,D, *p* > 0.05). Furthermore, ATFs had no significant effect on the organ index for the liver, kidney, and spleen (Figure 1E–G, *p* > 0.05). The findings indicate that ATFs did not exert an adverse effect on the mice’s food intake, water intake, and organ indices, thereby demonstrating a certain degree of safety.

### 2.2. ATFs Enhance Locomotor Activity in PD Mice

In this study, five distinct behavioral assessments were employed to ascertain the efficacy of ATFs in ameliorating motor dysfunction in PD mice. The rotenone-induced PD mice exhibited conspicuous movement disorders, as evidenced by a significantly prolonged time to complete the pole-climbing and sticker removal tasks (Figure 2A,B, *p* < 0.001), diminished limb muscle force (Figure 2C, *p* < 0.001), short time spent on the rotating rod (Figure 2D, *p* < 0.01), and reduced autonomous activity ability (Figure 2E–G, *p* < 0.001). In comparison with the ROT group, the ATFs could significantly reverse these effects (Figure 2A–G, *p* < 0.001). ATFs demonstrated a substantial amelioration of the rotenone-induced dyskinesia in the mice, and its effects on the results of multiple behavioral tests were comparable to or even superior to that of the positive control drug Madopar (MDR) (Figure 2A–C,F).

### 2.3. ATFs Reduce the Rotenone-Induced Loss of DA Neurons

Tyrosine hydroxylase (TH) is the rate-limiting enzyme in the synthesis of DA, and TH+ staining can be used to label DA neurons [6]. In this study, we observed a significant decrease in the number of DA neurons in the SN of the midbrain in the ROT group compared to the NCD group (*p* < 0.01). However, the ATF treatment resulted in a significant increase in the number of DA neurons in the SN (Figure 3A,B, *p* < 0.05). Furthermore, the Nissl staining results indicated that the nucleolus and Nissl bodies of neurons in the SN of the NCD group were dark blue and the cell structure was intact. In contrast, the Nissl bodies in the SN of the ROT group demonstrated a significant reduction, disintegration, or even disappearance. Following the ATF treatment, Nissl bodies appeared again in substantially higher numbers (Figure 3C,D, *p* < 0.01). This finding indicates that ATF treatment could significantly ameliorate the neuronal damage in the SN of mice.

### 2.4. ATFs Alleviate Rotenone-Induced Constipation Symptoms

Constipation is a prevalent GI symptom of PD [2]. In comparison with the CON group, the FBST in the ROT group demonstrated a significant increase (Figure 4A, *p* < 0.01), and the FN, FW, FWC, and GTR within a 6-h period was considerably reduced (Figure 4B–F, *p* < 0.05). These findings suggested that rotenone induced constipation symptoms in mice and that the administration of ATFs had a significant impact on reversing this effect. It was noteworthy that ATFs were even better than Madopar in promoting intestinal movement and improving fecal characteristics (Figure 4). These results suggest that ATFs could effectively be used to alleviate GI symptoms in PD mice.

### 2.5. ATFs Attenuate Rotenone-Induced Colonic Inflammation in PD Mice

Intestinal inflammation is a hallmark of PD in both human patients and animals [16]. The present study performed a pathological analysis of mouse colon tissues using H&E staining to reveal significant inflammatory cell infiltration into the colon tissue (Figure 5A), accompanied by a notable increase in the proportion of nuclei (Figure 5B, *p* < 0.05) and visible negative effects on the villus structure in the ROT group. Importantly, ATFs markedly reversed these effects. Furthermore, the relative expression levels of *TNF-α*, *IL-1β*, *IL-6*, *COX-2*, and *MCP-1* were significantly increased in the ROT group compared to the CON group (Figure 5C–G, *p* < 0.05), indicating that ROT successfully induced intestinal inflammation. Conversely, ATF administration significantly decreased the relative mRNA expression of *TNF-α*, *IL-1β*, *IL-6*, *COX-2*, and *MCP-1* in the colon tissue of rotenone-induced PD mice (Figure 5C–G, *p* < 0.05). These results suggest that ATFs can effectively alleviate inflammation in the colon tissue of PD mice.

### 2.6. ATFs Can Reverse the Rotenone-Induced Gut Barrier Damage

The intestinal barrier’s integrity is imperative for preventing the invasion of external pathogens and maintaining the stability of the intestinal environment. Previous studies have demonstrated that impairment of the intestinal barrier is closely associated with the occurrence of PD [21]. Muc-2 mucin is a principal constituent of intestinal epithelial tissue, while the Claudin, Occludin, and ZO families constitute the tight junctions between epithelial cells [22]. In this study, the mRNA expression levels of *Muc-2*, *Occludin*, *Claudin-3*, *Claudin-4*, and *ZO-1* were significantly downregulated in ROT mice compared with the CON group (*p* < 0.05), indicating compromised intestinal barrier function in these mice. Conversely, the administration of ATFs was found to prevent the impairment of intestinal barrier function, with the expression levels of *Muc-2*, *Occludin*, *Claudin-3*, *Claudin4*, and *ZO-1* being restored to levels comparable to those observed in the NCD group (Figure 6A–E, *p* < 0.05). This finding suggests that ATFs possess a notable capacity to mitigate the rotenone-induced damage to intestinal barrier function in PD mice.

### 2.7. ATFs Restore the Intestinal Microbiota Dysbiosis Caused by Rotenone

The 16S rRNA gene was sequenced in order to ascertain the effect of ATFs on PD mice’s gut microbiota, with the results illustrated in Figure 7. With the augmentation of both the sample size and the sequencing depth, the Sobs index in the rarefaction curves demonstrated a tendency to stabilize (Figure 7A), thus indicating that the sequencing depth was adequate to reflect the diversity of the samples and that the sequencing results are reliable. While rotenone and ATFs exhibited a modest influence on α-diversity (Figure 7B), the PCoA analysis revealed a substantial impact of rotenone on the β-diversity of the gut microbiota in the mouse cecal contents. Furthermore, the β-diversity of the gut microbiota in the ATF-treated mice was found to be similar to that of the CON group, suggesting that ATFs could exert a substantial effect on the intestinal microbial composition of rotenone-induced PD mice (Figure 7C).

An LEfSe analysis was utilized to identify the dominant species across multiple hierarchical levels among the different groups (Figure 7D). A total of 50 distinct taxa were identified across the three groups, comprising 1 phylum, 2 classes, 9 orders, 13 families, and 25 genera. Specifically, the CON, ROT, and ATF groups exhibited 15, 18, and 17 distinct levels of dominant microbial communities, respectively (Appendix A). The present study focused on the microbial communities that were significantly influenced by ROT and ATFs, particularly those that exhibited notable changes due to the ATF treatment.

At the phylum level, the administration of ROT resulted in a significant increase in the relative abundance of *Desulfobacterota*, while the ATF treatment led to a substantial decrease in the relative abundance of *Desulfobacterota* (Figure 7E, *p* < 0.05). Although no significant effect of ROT on *Actinobacteriota* was observed, the ATF treatment significantly enhanced the relative abundance of *Actinobacteriota* in PD mice (Figure 7E, *p* < 0.05). At the family level, ROT caused a significant increase in the relative abundance of the *Eubacterium_coprostanoligenes_group*, *Desulfovibrionaceae*, and *Streptococcaceae*, and a significant decrease in the relative abundance of *Rikenellaceae*, *Bifidobacteriaceae*, and UCG-010 (*p* < 0.05). However, the changes in these taxa were significantly reversed by the ATF treatment (Appendix A, *p* < 0.05). Although ROT had no significant effect on the *Eubacterium_coprostanoligenes_group*, *Acholeplasmataceae*, *Butyricicoccaceaeand*, and *norank_o__Izemoplasmatales*, the relative abundance of these taxa exhibited a significant increase following the administration of ATFs (Appendix A, *p* < 0.05).

At the genus level, ROT caused significant increases in the relative abundances of the *Lachnospiraceae_NK4A136_group*, *Desulfovibrio*, *norank_f__Erysipelotrichaceae*, *Blautia*, *Prevotellaceae*_*UCG-001*, the *Eubacterium_nodatum_group*, and *Intestinimonas*, and substantial reductions in the relative abundances of *Faecalibaculum*, the *Eubacterium_ruminantium_group*, *Alistipes*, *Bifidobacterium*, *norank_f__UCG-010*, *Parvibacter*, and *Anaeroplasma* (*p* < 0.05). However, this trend was significantly reversed after ATF treatment (*p* < 0.05). Furthermore, while no substantial effects of ROT on the *norank_f__Eubacterium_coprostanoligenes_group*, *UCG-009*, and *Paludicola* were observed, the relative abundances of these taxa were significantly increased after the ATF treatment (Figure 7F–H, *p* < 0.05). The above results demonstrated that ATFs can ameliorate the intestinal microbial disturbances induced by rotenone.

### 2.8. Correlation Analysis Between Specific Microbial Taxa and Key Phenotypes of PD Mice

In order to explore whether the gut microbiota plays an important role in the improvement of PD by ATFs, a bivariate correlation analysis was performed to explore the correlation between the microbial taxa significantly affected by ATFs and phenotypic indicators in PD mice. As illustrated in Figure 8A, the *Eubacterium_nodatum_group* exhibited a substantial correlation with all the motor phenotype indicators, including a notable positive association with PT and RST, and a significant negative correlation with GS, R-RT, SD, and SM (*p* < 0.05). Additionally, *norank_f_Erysipelotrichaceae*, the *Lachnospiraceae*_*NK4A136_group*, and *Desulfovibrio* were significantly positively correlated with PT and RST, and significantly negatively correlated with GS, SD, and SM (*p* < 0.05). In contrast, *Parvibacter*, the *norank_f_Eubacterium_coprostanoligenes_group*, and *UCG-009* were significantly positively correlated with SD and SM, and significantly negatively correlated with PT (*p* < 0.05). In addition, *norank*_*f_UCG-010*, *Bifidobacterium*, and *Faecalibaculum* were significantly positively correlated with SD and significantly negatively correlated with PT and RST; *Bifidobacterium* was also significantly positively correlated with GS and R-RT (*p* < 0.05). These findings imply that the Eubacterium_nodatum_group, *norank_f__Erysipelotrichaceae*, *Desulfovibrio*, and the *Lachnospiraceae*_*NK4A136_group* may play pivotal roles in PD motor dysfunction, while *Bifidobacterium*, *Faecalibaculum*, and *Parvibacter* may be crucial genera for ATFs to mitigate PD motor deficits.

Figure 8B illustrates that *Bifidobacterium* exhibited a significant negative correlation with FBST and a significant positive correlation with FWC, GTR, and FW (*p* < 0.05). *Faecalibaculum* also exhibited a significant negative correlation with FBST and a significant positive correlation with FWC (*p* < 0.05). Similarly, *Parvibacter* was significantly negatively correlated with FBST and positively correlated with FN and FW (*p* < 0.05). Furthermore, *norank_f__Erysipelotrichaceae* demonstrated significant correlations with all defecation phenotype indicators, including a significant positive correlation with FBST, while also showing significant negative correlations with FWC, GTR, FN, and FW (*p* < 0.05). In addition, *Desulfovibrio*, the *Lachnospiraceae*_*NK4A136*_*group*, *Prevotellaceae*_*UCG-001*, and *Intestinimonas* were found to be significantly positively correlated with FBST (*p* < 0.05). Concurrently, *Desulfovibrio* exhibited a significant negative correlation with FWC and GTR, while the *Lachnospiraceae*_*NK4A136*_*group* and *Prevotellaceae*_UCG-001 were significantly negatively correlated with FWC and FW (*p* < 0.05), respectively. These findings indicate that the *Eubacterium_nodatum_group*, *Desulfovibrio*, and the *Lachnospiraceae*_*NK4A136*_*group* play a crucial role in rotenone-induced GI dysfunction in PD, whereas *Bifidobacterium*, *Parvibacter*, and *Faecalibaculum* may represent key bacterial genera for alleviating constipation in PD.

## 3. Discussion

In the majority of patients diagnosed with PD, GI dysfunction, such as constipation, manifests earlier than the primary motor symptoms [23]. In this study, long-term low-dose rotenone was successfully used to model and simulate the dyskinesia and GI symptoms associated with PD. A recent study reported that the administration of rotenone for four weeks (30 mg/kg/day) partially reproduced GI lesions, including decreased fecal pellet production, and resulted in a reduction in TH-positive neurons in the SN [24]. In our current study, the rotenone treatment resulted in substantial GI dysfunction, including increased defecation time and decreased fecal number, fecal weight, fecal water content, and GI motility. Remarkably, the administration of ATFs significantly reversed these effects. Furthermore, the rotenone-treated mice exhibited motor impairments. These behavioral deficits were markedly improved by the ATF treatment, which fully confirmed their effects on PD GI dysfunction and motor symptoms.

The ATFs employed in this work included epicatechin, isoquercitrin, astragalin, kaempferol-3-O-rutinoside, procyanidin B2, and rutin. All of these flavonoid monomers have shown potential ameliorative effects on PD in previous studies. For instance, epicatechin-rich green tea polyphenols have been shown to offer some degree of protection to dopaminergic neurons in rat models of PD [25]. Isoquercitrin has been shown to exert a palliative effect on PD by enhancing the anti-apoptotic capacity of human proximal tubule epithelial cells (HK2 cells) [26]. Astragalin has been shown to counteract lipopolysaccharide-induced neuroinflammation in mice by reducing nitric oxide synthesis and pro-inflammatory cytokine production [27]. Both kaempferol and its derivatives have been shown to directly reduce the aggregation of α-syn and the production of Lewy bodies and to suppress central nervous system inflammation and oxidative stress [28]. Proanthocyanidin-rich extracts have been shown to activate cellular antioxidant mechanisms and alleviate mitochondrial dysfunction, consequently slowing down dopaminergic neuronal death induced by related toxins, which in turn mitigates PD symptoms [29]. Rutin has been demonstrated to exert neuroprotective effects through modulating mitochondrial autophagy by attenuating oxidative damage and depolarizing the mitochondrial membrane potential [30]. These previous studies corroborate the results of the present study. There are many kinds of compounds in ATFs, and the interactions between them are also complex. Therefore, it is of great significance to evaluate the overall effect of *Amomum tsaoko* total flavonoids on PD.

The degeneration and loss of DA neurons in the SN of the midbrain are the most prominent pathological features of PD [31]. The loss of DA neurons leads to a decrease in dopamine secretion, which is the main cause of motor dysfunction in PD patients [32]. TH is a biomarker of DA neurons and an enzyme necessary for dopamine synthesis. In rotenone-induced PD mice, studies have shown a significant reduction in the number of TH-positive cells and loss of DA nerve cells [33]. Icariin has been demonstrated to attenuate dopaminergic neuronal loss and motor impairment in rat models of PD [34]. Kaempferol-3-O-rutinoside-rich safflower flavonoid extract (SAFE) has also been shown to exert a similar effect to icariin [8]. In this study, we demonstrated that ATFs can significantly increase the number of TH-positive cells and Nissl-positive cells in the SN of rotenone-induced PD mice. These findings provide a solid foundation for further research to elucidate the mechanism through which ATFs improve PD.

A substantial body of evidence from human samples and animal models supports the involvement of intestinal inflammation in the development of PD [35]. The research has indicated that inflammatory factors in the gut can increase intestinal permeability, damage the blood–brain barrier, and accelerate the bidirectional transport of α-synuclein into the brain, thereby initiating α-synuclein misfolding in the brain [36]. Study has found that certain specific pro-inflammatory factors not only contribute to the onset of GI disease but also further induce brain inflammation and DA neuron death, ultimately leading to PD [37]. In addition, studies have suggested that inflammatory factors associated with chronic bowel disease may serve as risk factors for the development of PD [38]. The present study observed that the rotenone treatment exacerbated inflammatory cell infiltration into the colon of mice and significantly increased the expression of pro-inflammatory cytokines, suggesting that rotenone may trigger the secretion of specific immune cells that release pro-inflammatory factors. In addition, recent studies have reported that the neuroprotective effect of safflower flavonoid extract in the 6-OHDA-induced PD model may be related to its anti-inflammatory effect [39]. The potential of isoquercitrin to alleviate PD symptoms was attributed to its ability to inhibit p38-MAPK and NF-κB activation, thereby reducing the release of inflammatory factors and microglial hyperactivation [40]. In line with these findings, the present study demonstrated that ATFs can significantly inhibit the expression of pro-inflammatory cytokines. This finding indicates that the alleviation of PD symptoms by ATFs is closely related to their amelioration of intestinal inflammation.

Under normal conditions, the substantial mucus secreted by the intestinal epithelium and the robust intestinal barrier act to restrict the translocation of intestinal microorganisms. The innate immune system is responsible for protecting the GI tract from excessive inflammatory responses [41]. However, prolonged exposure to rotenone has been demonstrated to instigate persistent intestinal inflammation, culminating in severe dysbiosis of the intestinal microbiota [2,21]. These deleterious effects have been shown to amplify intestinal permeability by decreasing the expression of tight junction proteins [42]. Rutin, quercetin, and isoquercitrin have been found to strengthen the intestinal barrier by inhibiting the production of pro-inflammatory mediators as well as enhancing the production of tight junctions and mucins [43,44,45]. Consistent with the findings of previous studies, the present study found that ATFs significantly increased the expression of intestinal barrier factors in the colonic tissues of rotenone-induced PD mice.

Numerous studies have established that dysbiosis of the gut microbiota is present in both patients with PD and PD-like animal models [46,47]. In a mouse model of PD induced by the over-expression of α-Syn, the gut microbiota has been implicated in dyskinesia, microglial activation, and α-Syn-related pathophysiological processes. However, the administration of fecal microbiota from PD patients has been shown to exacerbate dyskinesia in mice overexpressing α-Syn [48]. This indicates that the gut microbiota contributes to the pathogenesis of PD. In this work, ATFs were capable of reversing the aberrant gut microbiota induced by rotenone. Correlations between specific microbial taxa and phenotypic indicators of motor defects and constipation in PD mice were observed. This suggests that these particular microbial taxa may play a crucial role in the therapeutic effects of ATFs in alleviating PD symptoms.

Previous clinical findings have demonstrated that the gut microbiota of PD patients contains an excess of the genus *Desulfovibrio* [49]. *Desulfovibrio* has been demonstrated to produce Fe_3_O_4_ and H_2_S, both of which have been shown to induce the oligomerization and aggregation of a-Syn [50,51]. In addition, partially hydrolyzed guar gum has been shown to alleviate constipation symptoms in mice by reducing *Desulfovibrio* [52]. In the present study, rotenone elevated the relative abundance of *Desulfovibrio* by more than 10-fold, whereas the ATF treatment significantly reduced its abundance. Of particular interest is the finding that the relative abundance of *Desulfovibrio* exhibited a strong correlation with mouse motor symptom phenotypes and defecation parameters. This finding suggests that the ameliorative effect of ATFs on rotenone-induced PD symptoms may be mediated, at least in part, by a reduction in the relative abundance of *Desulfovibrio*.

An increased abundance of *Prevotella* has been found to be associated with persistent inflammation in the gut and subsequent mucosal dysfunction and systemic inflammation [53]. Prior studies have indicated that rifaximin may exert anti-inflammatory and neuroprotective effects by impeding the surge in *Provotellaceae*_*UCG-001* abundance in PD mice [54]. In addition, aerobic exercise training has been shown to inhibit the relative abundance of *Prevotellaceae*_*UCG-001* in PD mice [55]. The present study also found that the relative abundance of *Provotellaceae*_*UCG-001* was significantly reduced by ATFs, and that the gene expression levels of pro-inflammatory factors and intestinal barrier-associated factors were significantly regulated. This finding suggests that the substantial decrease in *Provotellaceae*_*UCG-001* abundance may also play a crucial role in the alleviation of PD symptoms by ATFs.

*Bifidobacterium* is a well-established genus of probiotics that has demonstrated neuroprotective potential in a range of animal models [56,57]. A recently published study found that *Bifidobacterium* animalis ameliorated GI symptoms as well as motor symptoms, which was accompanied by alterations to the host gut microbiome in PD mice [58]. Prior research has indicated that specific flavonoid monomers, such as phlorizin, rutin, and proanthocyanidins, can enhance the relative abundance of *Bifidobacterium*, thereby mitigating colonic inflammation [59,60,61]. In the present study, ATFs also significantly increased the relative abundance of *Bifidobacterium*. Intriguingly, the correlation analysis revealed a strong association between *Bifidobacterium* and indicators of PD symptoms. This finding suggests that ATFs may alleviate PD by increasing the relative abundance of *Bifidobacterium*.

*Faecalibaculum* is a genus capable of producing short-chain fatty acids (SCFAs). A recent cohort study reported a significant decrease in the amount of fecal SCFAs in PD patients, and the level of fecal SCFAs was associated with the severity of the motor and cognitive dysfunctions and specific changes in pro-inflammatory bacteria in PD patients [62]. In addition, oral fullerene has been demonstrated to reverse the abnormal reduction in *Faecalibaculum* in MPTP-induced PD mice [63]. Furthermore, the administration of *Lactobacillus plantarum* has been shown to enhance the abundance of *Faecalibaculum* and to ameliorate motor deficits and constipation in PD mice [64]. In this study, rotenone administration led to a significant decrease in *Faecalibaculum*. However, the administration of ATFs resulted in a significant reversal of this trend. Concurrently, strong correlations were observed between specific microbial taxa and PD phenotypic indicators in mice. These findings suggest that the increase in *Faecalibaculum* abundance may play a crucial role in the alleviation of PD symptoms by ATFs.

Research has demonstrated an augmentation in the abundance of *Lachnospiraceae* in rotenone-stimulated mice [65], and a notable abundance of the *Lachnospiraceae*_*NK4A136*_*group* has been identified in constipated PD patients [66]. Furthermore, a significant increase in the abundance of *Erysipelotrichaceae* has been observed in mice induced with low-dose MPTP to model PD [67] and in primary PD patients [68]. Neohesperidin, a natural flavonoid extracted from citrus fruits, has been shown to significantly reduce the abundance of *Erysipelotrichaceae* in MPTP-injected mice [69]. In line with these findings, the present study demonstrated that ATFs significantly reversed the substantial increase in the abundance of *Lachnospiraceae*, the *Lachnospiraceae*_*NK4A136*_*group*, *Erysipelotrichaceae*, and *norank_f__Erysipelotrichacea* induced by rotenone. This finding suggests that the ameliorative effect of ATFs on rotenone-induced PD symptoms may be achieved, at least in part, by reducing the relative abundance of the *Lachnospiraceae*_*NK4A136*_*group* and *norank_f__Erysipelotrichaceae*. The results of our correlation analysis further corroborate this speculation.

It is noteworthy that the ATF treatment led to a substantial increase in several low-abundance taxa, including the *norank__f_Eubacterium_coprostanoligenes_group* and *Parvibacter*, which have been strongly linked to motor and constipation symptoms [70]. It was found that black goji anthocyanins increased the relative abundance of the *norank_f_Eubacterium_coprostanoligenes_group* and *Parvibacter* in MPTP-induced PD mice. Shouhui tongbian capsules have been observed to ameliorate intestinal pathological changes and promote intestinal motility in mice by modulating the equilibrium of pivotal small intestinal mucosal bacteria, including *Parvibacter* [71]. It is also notable that ATFs have been observed to reduce the relative abundance of the *Eubacterium*_*nodatum*_*group*, which has been identified as a significant indicator of PD periodontitis in patients [72]. Furthermore, the correlation analysis revealed a significant correlation between the *Eubacterium*_*nodatum*_*group* and all indicators of the motor phenotypes. In summary, the gut microbiota may play important roles in the alleviation of motor symptoms and GI dysfunction in PD mice by ATFs. Several noteworthy conditionally pathogenic bacteria (*Desulfovibrio*, *Provotellaceae_UCG-001*, the *Lachnospiraceae*_*NK4A136_group*, *norank_f__Erysipelotrichacea*, and the *Eubacterium*_*nodatum*_*group*) and probiotics (*Bifidobacterium* and *Faecalibaculum*) might be the pivotal factors.

## 4. Materials and Methods

### 4.1. Primary Materials and Reagents

The reagents used were rotenone (ROT), carboxymethylcellulose sodium (CMC-Na, Sigma Aldrich Trading Co., Ltd., Shanghai, China), Madopar (Shanghai Roche Pharmaceutical Co., Ltd., Shanghai, China), PCR primers (Shanghai Generay Bioengineering Co., Ltd., Shanghai, China), and a monoclonal anti-tyrosine hydroxylase (TH) antibody (Abcam Inc., Milpitas, CA, USA).

### 4.2. ATF Preparation

The provenance of the AT fruits was Gongshan County of Yunnan Province. The ATFs were prepared in accordance with the published method from our group [73]. The AT was pulverized and then extracted using a sonicator. Subsequently, the upper sample solution was prepared at a specific concentration and pH. Adsorption and desorption were conducted with HPD300 macroporous resin(Beijing Solarbio Science & Technology Co., Ltd., Beijing, China), and the 20% and 30% ethanol eluates were concentrated and vacuum-dried, and stored. The 20% and 30% ethanol-eluting fractions contained more than 90% flavonoid compounds (based on the NaNO_3_-Al (NO_3_)_3_-NaOH colorimetric method) with a relative abundance of 73.83% (based on plant-wide targeted metabolomics). The components of the ATFs were determined using UPLC-QTRAP-MS/MS [73]. The ATFs obtained from the preparation primarily consisted of (+)-epicatechin, isoquercitrin, astragalin, kaempferol-3-O-rutinoside, procyanidin B2, vanillin, rutin, and other compounds (Appendix A).

### 4.3. Animals and Experimental Design

A total of 48 specific pathogen-free (SPF) grade, 8-week-old, male C57BL/6 mice (20–22 g) were purchased from Beijing Vital River Laboratory Animal Technology Co., Ltd. (Beijing, China). The animals were housed in the Yunnan Key Laboratory of Precision Nutrition and Personalized Food Manufacturing. The temperature and relative humidity were maintained at 23 ± 2 °C and 55–65% under a 12-h light/dark cycle, with free access to food and water. All the procedures were reviewed and approved by the Life Science Ethics Committee of Yunnan Agricultural University (Approval No. 202312008).

Following a week of acclimatization, the mice were randomly divided into four groups (*n* = 12), namely the normal chow diet (NCD) group, the rotenone gavage (ROT) group, the positive control Madopar (MDR) group, and the *Amomum tsaoko* flavonoid (ATF) group, with four mice in each cage. The scheme for the animal experiments is illustrated in Figure 1A. Briefly, the NCD group was administered sterilized ultrapure water (UP water). The ROT group was given 30 mg/kg·BW of rotenone, the MDR group was given 30 mg/kg·BW of rotenone and 50 mg/kg·BW of Madopar, and the ATF group was given 30 mg/kg·BW of rotenone and 50 mg/kg·BW of ATFs. During the ATF treatment, the oral gavage was administered once a day for 8 weeks.

### 4.4. Behavioral Evaluation Tests

Five behavioral tests were performed in order to evaluate the mice’s motor function. All the behavioral equipment was purchased from SANS Biotechnology Co., Ltd. (Nanjing, China). In all the behavioral tests, the mice were subjected to three trials, with an interval of one hour between trials.

#### 4.4.1. Pole Test

The pole-climbing device (SA111, SANS, Nanjing, China) had a height of 50 cm, a diameter of 1 cm, and was topped with a 35 mm diameter sphere. The climbing pole was angled at 45°, and during the test, the mice were placed on the sphere. The time it took for the mice to climb to the bottom of the pole was recorded. Three trials were performed for each mouse [16].

#### 4.4.2. Adhesive Removal Test

Each mouse was placed in a clean cage and allowed one minute of free movement to acclimatize. At the commencement of the formal experiment, a circular adhesive label (0.3 cm × 0.4 cm) was gently affixed to the forepaw of the mouse, and the time for complete removal of the label was recorded [2].

#### 4.4.3. Grip Strength Test

A laboratory grip strength meter (SA415, SANS, Nanjing, China) was positioned on a stable operating table and a mouse was placed on the meter shelf. Once its limbs grasped the shelf, the tail of the mouse was gently pulled in the horizontal direction until the mouse released the shelf. The maximum grasping force of the mouse was recorded [2].

#### 4.4.4. Rota-Rod Test

A rota-rod treadmill (SA120M, SANS, Nanjing, China) was set at a speed of 35 r/min and test time of 180 s. The mice were placed on the rota-rod for 30 s prior to the commencement of the experiment. The time for the mice to fall off the rotating rod was recorded [16].

#### 4.4.5. Open Field Test

The experiment utilized opaque Plexiglass (SA215, SANS, Nanjing, China) with a bottom with dimensions of 100 cm × 100 cm and a height of 40 cm. At the onset of the experiment, the mice were positioned at any location in the open field. The statistical software automatically recorded the moving distance and average speed of the rats in a 5 min period, while the camera tracked and depicted the movement trajectory of the mice. The open-field experimental box was cleaned before each test [74].

### 4.5. Defecation Test

The night prior to the observation of defecation, the mice in each group were treated with food withdrawal. After 12 h of fasting, mice in the different groups were gavaged normally; 30 min later, the mice were gavaged with self-made ink, and each mouse was individually placed in a cage with freely available water. The first black stool time (FBST) was recorded, and the fecal number (FN) and fecal weight (FW) of the mice were determined within 6 h. At the end of the observation, the collected fecal samples were dried at 80 °C until a constant weight was reached, which was used to calculate the fecal water content (FWC).

### 4.6. Gastrointestinal Transit Test

The evening before the mice were euthanized, all the mice were treated with food deprivation for 12 h. The next morning, the mice in the different groups were given normal gavage; after 30 min, ink was gavaged into the stomach as an indicator of GI movements. Following a 20 min period, the mice were euthanized. The abdominal cavity was then opened immediately, the intestines were removed, and the length of the small intestine, as well as the distance moved by the ink, was measured. The gastrointestinal transit rate (GTR) was calculated.

### 4.7. Immunohistochemistry

Following dissection, fixation, dehydration, and paraffin embedding, the brain tissues were cut into 4 μm thick slices. These slices were dewaxed and rehydrated in xylene (Sinopharm Chemical Reagent Co., Ltd. Shanghai, China), treated with microwave energy in an EDTA repair solution, and the endogenous peroxidase activity was inhibited with 3% H_2_O_2_. The tissues were then incubated with bovine serum albumin to block non-specific antibody binding, followed by incubation in primary anti-TH antibody (1:500, ab137869, Abcam) overnight at 4 °C. Biotinylated secondary antibody (1:200, GB23303, Servicebio, Wuhan, China) was subsequently added for 30 min at 37 °C, followed by DAB staining and hematoxylin re-staining. The resulting microphotographs were collected using an inverted microscope (Nikon Ci-S, Tokyo, Japan) microscope and CellSens Entry(Nikon DS-U3, Tokyo, Japan). Positive cells were counted using Image Pro Plus 6.0 software. Each section was assessed at six randomly selected fields.

### 4.8. Nissl Staining

The mouse brains were cut into 4 μm slices and attached to a slide. Xylene was used for dewaxing, followed by a gradient treatment with 100%, 95%, and 80% ethanol until the sample was hydrated. The sections were then placed in 0.5% toluidine blue, stained at 56 °C for 3 min, and then rinsed with water. After that, 95% and 100% ethanol solutions were utilized for the dehydration treatment, followed by xylene to make the sections transparent, which were finally sealed with neutral gum. Images were collected using an Olympus CX43 microscope and CellSens Entry, and the number of Nissl-positive cells was counted using Image Pro Plus 6.0 software. Each slice was analyzed at six randomly selected fields.

### 4.9. HE Staining of Colon Tissue

The colon tissue was fixed in a 4% paraformaldehyde solution, embedded, and cut into 3 µm thick sections. The sections were sequentially deparaffinized with xylene, hydrated with graded concentrations of ethanol, rinsed with distilled water, stained with hematoxylin for 15 min, differentiated with 0.5% hydrochloric acid in an ethanol solution for 30 s, rinsed with distilled water, stained with 0.5% eosin for 10 min, rinsed with distilled water, processed with xylene, and finally sealed with a neutral resin. Images of each section were collected using an Olympus CX43 microscope and CellSens Entry.

### 4.10. Total RNA Extraction and Quantitative PCR Analysis of Gene Expression

According to the manufacturer’s instructions, total RNA was extracted from colon tissue using a bead-beating method and the TaKaRa MiniBEST Universal RNA Extraction Kit (9767, Takara, Kyoto, Japan). The RNA concentration and purity were subsequently determined using a spectrophotometer. cDNA was synthesized using the PrimeScript TM RT reagent Kit and gDNA Eraser Kit (RR047A, Takara, Kyoto, Japan). The quantitative PCR reaction system was configured using the TB Green^®^ Premix Ex TaqTM II kit (RR820A, Takara, Kyoto, Japan). The PCR reaction conditions were performed using a previously described method [20]. β-actin was utilized as the internal reference gene to calibrate the relative expression of the target gene mRNA. The relative expression of the target gene was then calculated using the 2^−∆∆CT^ method. The sequences of the primers for the target gene and housekeeping gene are presented in Appendix A.

### 4.11. DNA Extraction and 16S rRNA Gene Sequencing of Cecal Contents

The total genomic DNA of the microbial community in the cecal contents and fecal samples was extracted using the E.Z.N.A.^®^ Soil DNA Kit (Omega Bio-tek, Norcross, GA, USA) according to the manufacturer’s instructions. The quality of the extracted genomic DNA was determined using 1% agarose gel electrophoresis, and the DNA concentration and purity were determined using a NanoDrop2000 (Thermo Scientific Inc., Shanghai, China). Using the extracted DNA as the template, the upstream primer 338F and the downstream primer 806R carrying the barcode sequence were used to amplify the V3–V4 variable region. The PCR conditions and protocol for the purification of the PCR products were described previously [13]. The purified amplicons were mixed in equal molar amounts and subjected to double-end sequencing on the Illumina PE300 platform (Illumina, San Diego, CA, USA) in accordance with the standard protocol of Major Biomedical Technology Co., Ltd. (Shanghai, China). The bioinformatics analysis was conducted on Majorbio’s cloud platform. The original sequencing reads were stored in the NCBISequence Reader Archive (SRA) database under BioProject accession number PRJNA1209613.

### 4.12. Data Analysis

The data were analyzed using GraphPad Prism 10.1.2 software, and the results are presented as the mean ± standard error of the mean (mean ± SEM). To identify significant differences between three or more groups, one-way ANOVA (for one variable) or two-way ANOVA (for two variables) was performed, followed by a post hoc test (Student–Newman–Keuls comparison test) unless otherwise stated in the figure legends. Multivariate analyses, i.e., principal coordinate analysis (PCoA), and LEfSe analyses were carried out using the Majorbio cloud platform (http://www.majorbiogroup.com/, accessed on 3 November 2024). Bivariate correlations between the host parameters and gut microbes were determined using a cloud platform (https://www.omicstudio.cn/, accessed on 16 December 2024). Unless stated otherwise, *p* < 0.05 was considered statistically significant.

## 5. Conclusions

This study presents the novel finding that ATFs exert a protective effect against low-dose rotenone-induced PD in mice, including the amelioration of motor deficits and constipation symptoms, the preservation of DA neurons, a reduction in inflammatory responses, and the enhancement of the gut barrier. Notably, ATFs significantly influenced the structure and composition of the gut microbiota in PD mice. The correlation analysis between the gut microbiota and the PD phenotypic indicators provides important evidence for elucidating the possible mediating role of the gut microbiota in the amelioration of PD by ATFs. Several conditionally pathogenic and probiotic taxa may play an important role in the alleviation of PD by ATFs. In conclusion, the oral administration of ATFs may have potential as a modality for the prevention and treatment of PD. The anti-PD mechanism of ATFs may be related to the regulation of the gut microbiota; however, further in-depth investigations are required to determine the molecular mechanisms. In addition, the investigation and confirmation of the main active components of ATFs against PD are also of interest. The results of this study provide a theoretical basis for the future application of ATFs in clinical research, food products, and medicine.

## Figures and Tables

**Figure 1 ijms-26-01676-f001:**
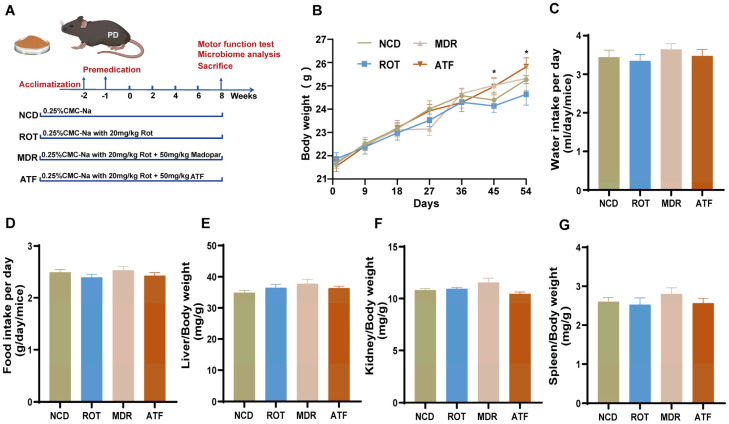
Effects of ATFs on body weight, food and water intake, and organ indices in mice. (**A**) Flow chart for the animal treatment protocol; (**B**) body weights of mice; (**C**) water intake; (**D**) food intake; (**E**–**G**). Liver, kidney, and spleen indices. All data are presented as the mean ± SEM, *n* = 12. * compared with the ROT group. * *p* < 0.05.

**Figure 2 ijms-26-01676-f002:**
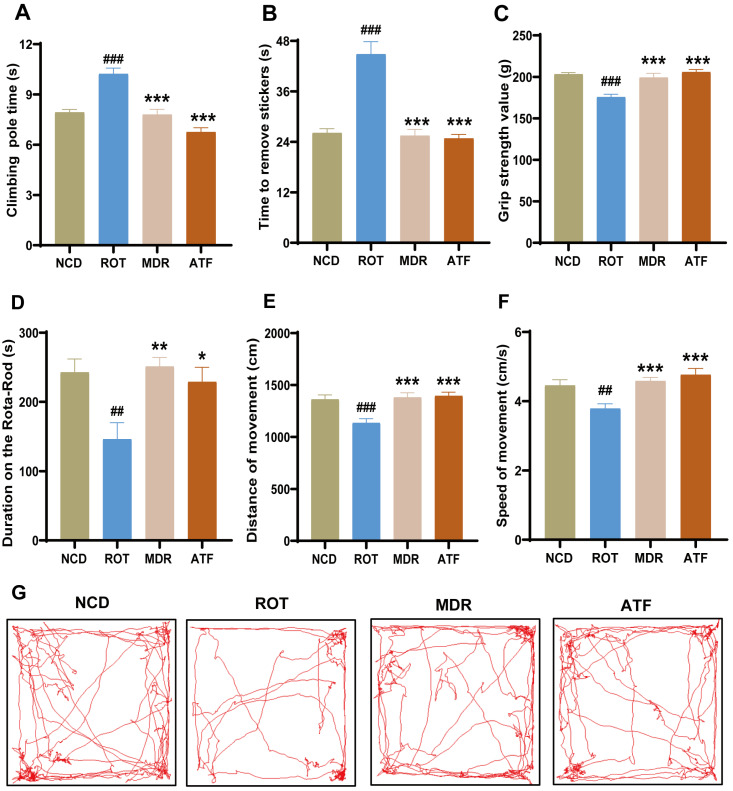
The effect of ATFs on the motor behavior of rotenone-induced PD mice. (**A**) Pole test (PT); (**B**) adhesive removal test (RST); (**C**) grip strength test (GS); (**D**) rota-rod test (R-RT); (**E**,**F**) open field test, including movement distance (SM) and movement speed (SD) measurements; (**G**) representative activity trajectory images in open field test. Data are presented as the mean ± SEM, *n* = 12. # compared with the NCD group; * compared with the ROT group. ## *p* < 0.01, ### *p* < 0.001; * *p* < 0.05, ** *p* < 0.01, *** *p* < 0.001.

**Figure 3 ijms-26-01676-f003:**
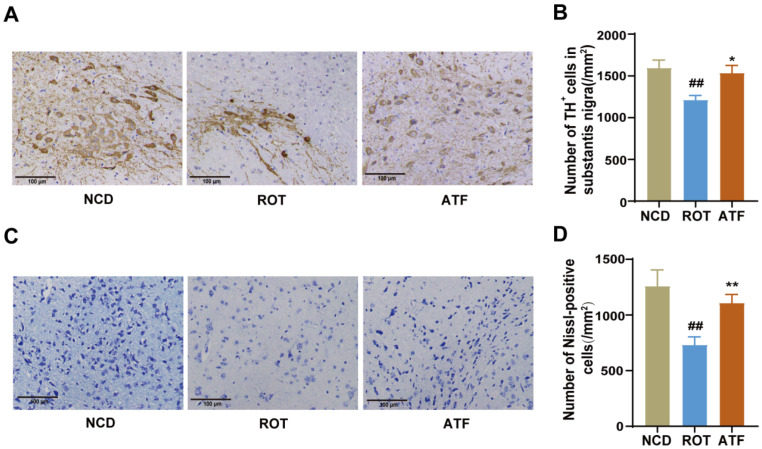
The effect of ATFs on the number of dopaminergic neurons in PD mice. (**A**,**B**) Representative images (**A**) and quantitative analysis of the optical density (**B**) of tyrosine hydroxylase (TH) immunohistostaining in the substantia nigra (SN); (**C**) representative images of Nissl staining in the SN; (**D**) Comparison of the number of Nissl-positive neurons in the SN of the mice in each group. Scale bar is 100 μm. Data are presented as the mean ± SEM, *n* = 8. # compared with the CON group; * compared with the ROT group. ## *p* < 0.01; * *p* < 0.05, ** *p* < 0.01.

**Figure 4 ijms-26-01676-f004:**
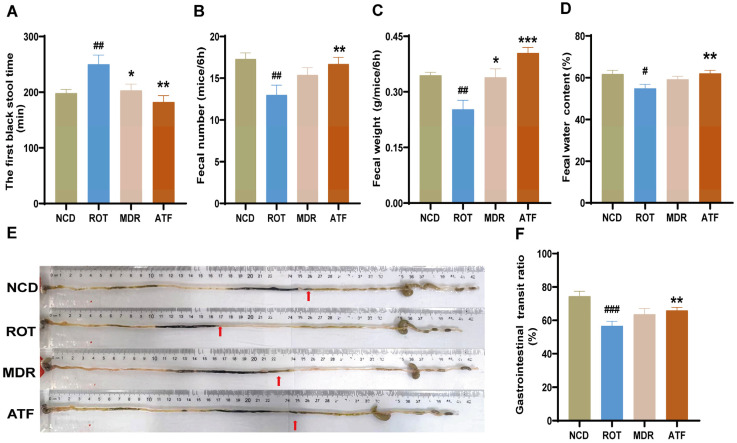
The effect of ATFs on constipation symptoms in PD mice. (**A**) The first black stool time (FBST); (**B**) fecal number (FN) in 6 h; (**C**) fecal wet weight (FW) in 6 h; (**D**) fecal water content (FWC); (**E**) representative pictures of ink propulsion distance in each group of mice; (**F**) gastrointestinal transit rate (GTR). Data are presented as the mean ± SEM, *n* = 12. # compared with the CON group; * compared with the ROT group. # *p* < 0.05, ## *p* < 0.01, ### *p* < 0.001; * *p* < 0.05, ** *p* < 0.01, *** *p* < 0.001.

**Figure 5 ijms-26-01676-f005:**
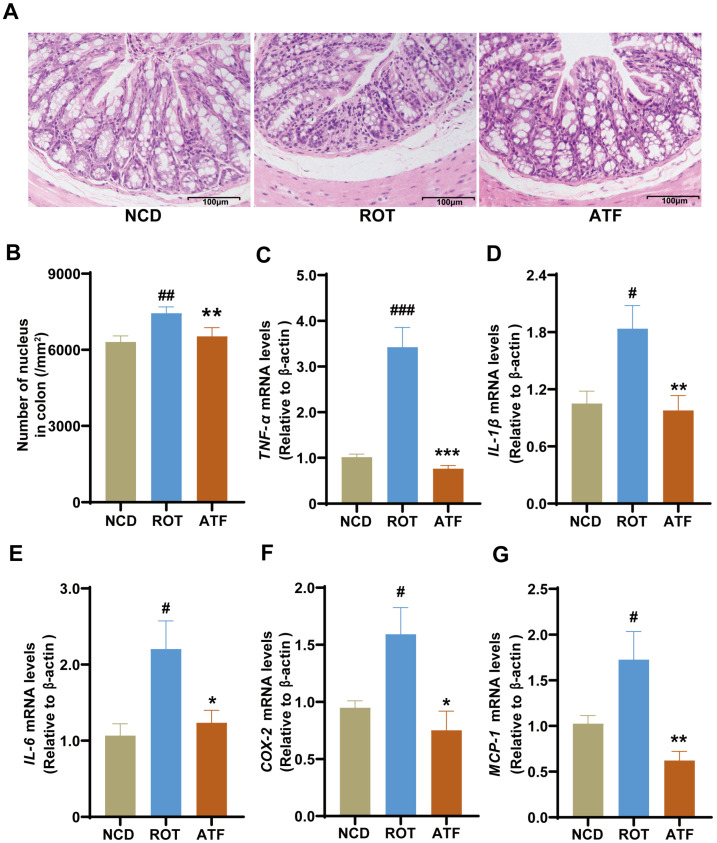
The effect of ATFs on inflammatory factors in PD mice. (**A**) Representative images of colon sections stained with H&E; (**B**) quantitative analysis of the optical density of nuclei in the colon; (**C**–**G**) mRNA expression of *TNF-α*, *IL-1β*, *IL-6*, *COX-2*, and *MCP-1* in the colon. Data are presented as the mean ± SEM, *n* = 8. # compared with the CON group; * compared with the ROT group. # *p* < 0.05, ## *p* < 0.01, ### *p* < 0.001; * *p* < 0.05, ** *p* < 0.01, *** *p* < 0.001.

**Figure 6 ijms-26-01676-f006:**
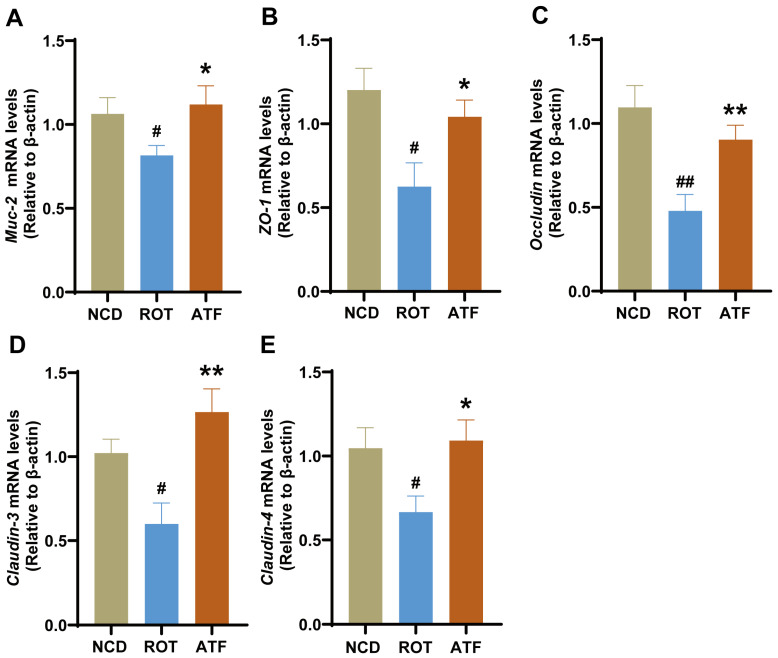
Effect of ATFs on intestinal barrier-related factors in PD mice. The mRNA expression of (**A**) *Muc-2*, (**B**) *ZO-1*, (**C**) *Occludin*, (**D**) *Claudin-3*, and (**E**) *Claudin-4* in the colon. Data are presented as the mean ± SEM, *n* = 8. # compared with the CON group; * compared with the ROT group. # *p* < 0.05, ## *p* < 0.01; * *p* < 0.05, ** *p* < 0.01.

**Figure 7 ijms-26-01676-f007:**
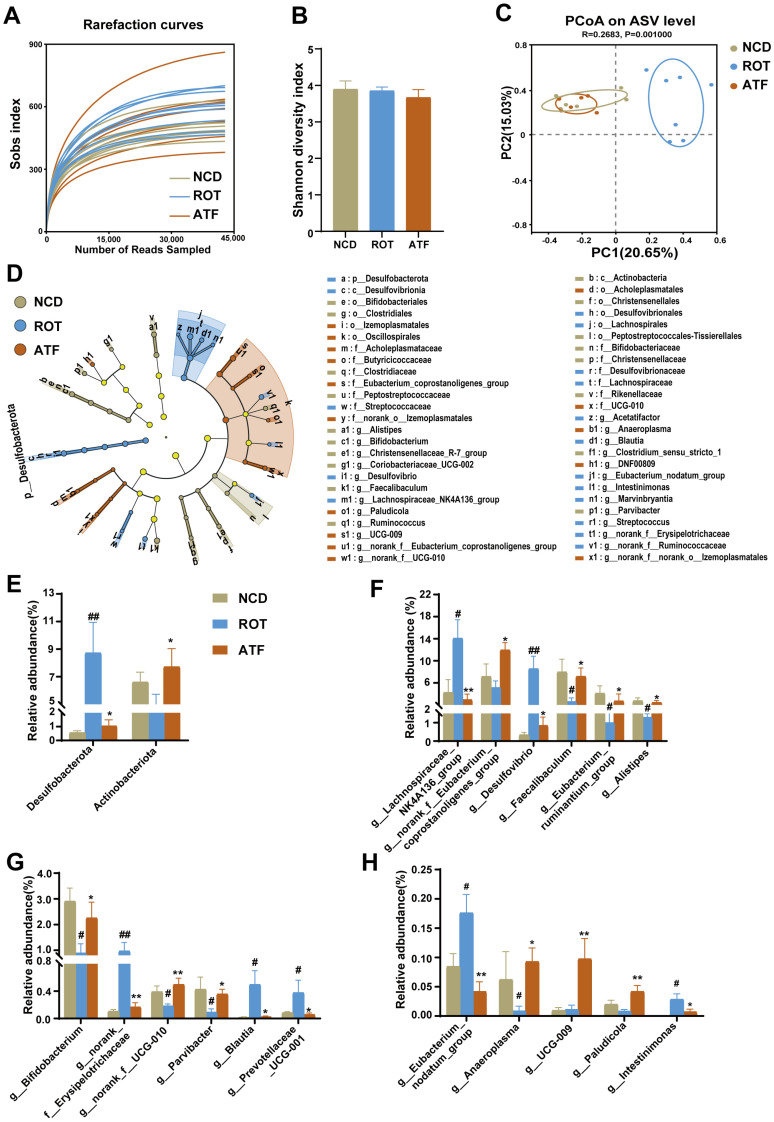
ATFs reversed the gut microbial community structure and composition in PD mice. (**A**) Rarefaction curves; (**B**) α-diversity index; (**C**) PCoA based on the Hellinger distance algorithm; (**D**) linear discriminant analysis effect size (LEfSe) (LDA score > 2.0); (**E**) relative abundance of microbes at the phylum level; (**F**–**H**) relative abundances of microbes at the genus levels that were significantly affected by ROT or ATFs. All data are presented as the mean ± SEM, *n* = 6–7. # compared with the CON group; # compared with the CON group; * compared with the ROT group. # *p* < 0.05, ## *p* < 0.01; * *p* < 0.05, ** *p* < 0.01.

**Figure 8 ijms-26-01676-f008:**
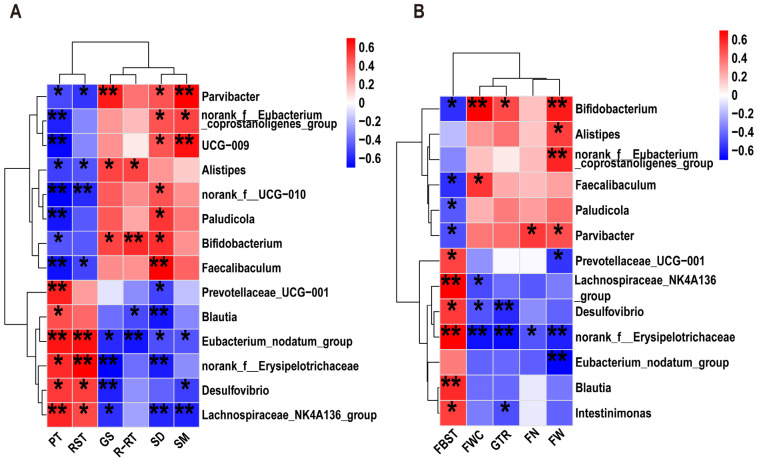
Correlation analysis between phenotypic indices and genus-level differences in microbial communities. (**A**) Correlations between genus-level differential microbes and motility phenotype parameters; (**B**) correlations between differential genera and defecation parameters. The color at each point of intersection indicates the value of the r coefficient (*n* = 8). Benjamini–Hochberg (BH) procedures were used to adjust the *p* values for multiple testing. * Indicates that there is a significant correlation between these two parameters (*p* < 0.05). * *p* < 0.05, ** *p* < 0.01. Climbing pole time: PT; grip strength value: GS; duration on the rota-rod: R-RT; time to remove stickers: RST; distance of movement: SD; speed of movement: SM; first black stool time: FBST; fecal water content: FWC; fecal wet weight: FW; fecal number: FN.

## Data Availability

The raw reads of the 16S rRNA gene sequence data were deposited into the NCBI Sequence Read Archive (SRA) database under BioProject accession number PRJNA1209613.

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
