# Peer review of "Correlations Between Amelioration of Rotenone-Induced Parkinson’s Symptoms by Amomum tsaoko Flavonoids and Gut Microbiota in Mice"

_ijms, 2025, doi:10.3390/ijms26041676_

Round 1
Reviewer 1 Report
Comments and Suggestions for Authors
Dear Authors,
Please find the comments below.
1. In the abstract, the abbreviation (AT) should be added to the Amomum tsaoko when it is introduced for the first time.
2. Although Amomum tsaoko has been demonstrated to have anti-inflammatory effects and influence gut microbiota composition, there is a lack of a direct connection between these findings and Parkinson’s disease. In order to establish a more solid link between the study results and the impact on Parkinson’s disease, it is necessary to further discuss previous studies that have examined changes in biomarker levels, gut microbiota composition, and the causal relationship between key flavonoids in AT and Parkinson’s disease.
Author Response
Response to comments of Reviewer #1
Dear Authors, Please find the comments below.
- In the abstract, the abbreviation (AT) should be added to the Amomum tsaoko when it is introduced for the first time.
Response: Thank you for your careful review. We have corrected this information in the abstract.
Abstract: Parkinson's disease (PD) is the second most common neurodegenerative disease, but the existing therapeutic drugs for PD have limitations; thus, there is an urgent need to discover new methods of prevention and treatment. Amomum tsaoko Crevost et Lemarie (AT) is a classic Traditional Chinese Medicine and food; its main pharmacological effect is the regulation of the gastrointestinal tract.
- Although Amomum tsaoko has been demonstrated to have anti-inflammatory effects and influence gut microbiota composition, there is a lack of a direct connection between these findings and Parkinson’s disease. In order to establish a more solid link between the study results and the impact on Parkinson’s disease, it is necessary to further discuss previous studies that have examined changes in biomarker levels, gut microbiota composition, and the causal relationship between key flavonoids in AT and Parkinson’s disease.
Response: Thank you very much for your valuable suggestions for our research. In this study, it was found that AT and ATFs have anti-inflammatory and gut microbiota-regulating effects through searching literature. However, so far no one has studied the relationship between AT and ATFs and Parkinson's disease. This study is the first to explore and find that ATFs can improve Parkinson's disease, inhibit rotenone-induced inflammation, and improve gut microbiota disorder, thus establishing the relationship between them. We have carried out relevant discussions in the manuscript.
The first is a discussion of the changes in the biomarker levels:
The degeneration and loss of DA neurons in the SN of the midbrain are the most prominent pathological features of PD [31]. The loss of DA neurons leads to a decrease in dopamine secretion, which is the main cause of motor dysfunction in PD patients [32]. TH is a biomarker of DA neurons and an enzyme necessary for dopamine synthesis. In rotenone-induced PD mice, studies have shown a significant reduction in the number of TH-positive cells and loss of DA nerve cells [33]. Icariin has been demonstrated to attenuate dopaminergic neuronal loss and motor impairment in rat models of PD [34]. Kaempferol-3-O-rutinoside-rich safflower flavonoid extract (SAFE) has also been shown to exert a similar effect to icariin [8]. In this study, we demonstrated that ATFs can significantly increase the number of TH-positive cells and Nissl-positive cells in the SN of rotenone-induced PD mice. These findings provide a solid foundation for further research to elucidate the mechanism through which ATFs improve PD.
The second is a discussion on the improvement of gut microbiota composition by ATFs:
Numerous studies have established that dysbiosis of the gut microbiota is present in both patients with PD and PD-like animal models [46,47]. In a mouse model of PD induced by the over-expression of α-Syn, the gut microbiota has been implicated in dyskinesia, microglial activation, and α-Syn-related pathophysiological processes. However, the administration of fecal microbiota from PD patients has been shown to exacerbate dyskinesia in mice overexpressing α-Syn [48]. This indicates that the gut microbiota contributes to the pathogenesis of PD. In this work, ATFs were capable of reversing the aberrant gut microbiota induced by rotenone. Correlations between specific microbial taxa and phenotypic indicators of motor defects and constipation in PD mice were observed. This suggests that these particular microbial taxa may play a crucial role in the therapeutic effects of ATFs in alleviating PD symptoms.
Although there are no reports on the causal relationship between the whole flavonoids in AT and Parkinson's disease, we discuss the effects of these compounds on Parkinson's disease in other plants or as single compounds in the manuscript.
The ATFs employed in this work included epicatechin, isoquercitrin, astragalin, kaempferol-3-O-rutinoside, procyanidin B2, and rutin. All of these flavonoid monomers have shown potential ameliorative effects on PD in previous studies. For instance, epicatechin-rich green tea polyphenols have been shown to offer some degree of protection to dopaminergic neurons in rat models of PD [25]. Isoquercitrin has been shown to exert a palliative effect on PD by enhancing the anti-apoptotic capacity of human proximal tubule epithelial cells (HK2 cells) [26]. Astragalin has been shown to counteract lipopolysaccharide-induced neuroinflammation in mice by reducing nitric oxide synthesis and pro-inflammatory cytokine production [27]. Both kaempferol and its derivatives have been shown to directly reduce the aggregation of α-syn and the production of Lewy bodies and to suppress central nervous system inflammation and oxidative stress [28]. Proanthocyanidin-rich extracts have been shown to activate cellular antioxidant mechanisms and alleviate mitochondrial dysfunction, consequently slowing down dopaminergic neuronal death induced by related toxins, which in turn mitigates PD symptoms [29]. Rutin has been demonstrated to exert neuroprotective effects through modulating mitochondrial autophagy by attenuating oxidative damage and depolarizing the mitochondrial membrane potential [30]. These previous studies corroborate the results of the present study. There are many kinds of compounds in ATFs, and the interactions between them are also complex. Therefore, it is of great significance to evaluate the overall effect of Amomum tsaoko total flavonoids on PD.
Finally, thank you again for your professional advice.
References in this response:
- Zakaryan, H.; Arabyan, E.; Oo, A.; Zandi, K. Flavonoids: promising natural compounds against viral infections. Arch. Virol. 2017, 162, 2539-2551.
- Guo, S.; Yan, J.; Yang, T.; Yang, X.; Bezard, E.; Zhao, B. Protective effects of green tea polyphenols in the 6-OHDA rat model of Parkinson's disease through inhibition of ROS-NO pathway. Biol. Psychiatry2007, 62, 1353-1362.
- Magalingam, K.B.; Radhakrishnan, A.; Ramdas, P.; Haleagrahara, N. Quercetin glycosides induced neuroprotection by changes in the gene expression in a cellular model of Parkinson's disease. J. Mol. Neurosci.2015, 55, 609-617.
- Kim, E.H.; Shim, Y.Y.; Lee, H.I.; Lee, S.; Reaney, M.J.T.; Chung, M.J. Astragalin and Isoquercitrin Isolated from Aster scaber Suppress LPS-Induced Neuroinflammatory Responses in Microglia and Mice. Foods2022, 11.
- Jin, S.; Zhang, L.; Wang, L. Kaempferol, a potential neuroprotective agent in neurodegenerative diseases: From chemistry to medicine. Biomed. Pharmacother.2023, 165, 115215.
- Tambe, M.A.; de Rus Jacquet, A.; Strathearn, K.E.; Hensel, J.A.; Colon, B.D.; Chandran, A.; Yousef, G.G.; Grace, M.H.; Ferruzzi, M.G.; Wu, Q. et al. Protective Effects of Polyphenol-Rich Extracts against Neurotoxicity Elicited by Paraquat or Rotenone in Cellular Models of Parkinson's Disease. Antioxidants2023, 12.
- Lai, X.; Zhang, Y.; Wu, J.; Shen, M.; Yin, S.; Yan, J. Rutin Attenuates Oxidative Stress Via PHB2-Mediated Mitophagy in MPP(+)-Induced SH-SY5Y Cells. Neurotox. Res.2023, 41, 242-255, doi:10.1007/s12640-023-00636-5.
- Zhao, M.; Wang, B.; Zhang, C.; Su, Z.; Guo, B.; Zhao, Y.; Zheng, R. The DJ1-Nrf2-STING axis mediates the neuroprotective effects of Withaferin A in Parkinson's disease. Cell Death Differ.2021, 28, 2517-2535.
- Johnson, M.E.; Salvatore, M.F.; Maiolo, S.A.; Bobrovskaya, L. Tyrosine hydroxylase as a sentinel for central and peripheral tissue responses in Parkinson's progression: Evidence from clinical studies and neurotoxin models. Prog. Neurobiol.2018, 165-167, 1-25.
- Cramb, K.M.L.; Beccano-Kelly, D.; Cragg, S.J.; Wade-Martins, R. Impaired dopamine release in Parkinson's disease. Brain.2023, 146, 3117-3132.
- Lu, D.; Chen, C.; Zheng, Y.; Li, D.; Wang, G.; Liu, J.; Shi, J.; Zhang, F. Combination Treatment of Icariin and L-DOPA Against 6-OHDA-Lesioned Dopamine Neurotoxicity. Front. Molec. Neurosci.2018, 11, 155.
- Romano, S.; Savva, G.M.; Bedarf, J.R.; Charles, I.G.; Hildebrand, F.; Narbad, A. Meta-analysis of the Parkinson's disease gut microbiome suggests alterations linked to intestinal inflammation. npj Parkinsons Dis.2021, 7, 27.
- Sun, M.; Shen, Y. Dysbiosis of gut microbiota and microbial metabolites in Parkinson's Disease. Ageing Res. Rev.2018, 45, 53-61.
- Morais, L.H.; Boktor, J.C.; MahmoudianDehkordi, S.; Kaddurah-Daouk, R.; Mazmanian, S.K. alpha-Synuclein Overexpression and the Microbiome Shape the Gut and Brain Metabolome in Mice. bioRxiv2024.
Reviewer 2 Report
Comments and Suggestions for Authors
The manuscript "Correlations between amelioration of rotenone-induced Park-inson's symptoms by Amomum tsaoko flavonoids and gut microbiota in mice" by Liu et al. reports the use of an Amomum tsaoko extract, assessed for composition in a previous publication (https://doi.org/10.1016/j.fochx.2025.102177), on rotenone-induced Parkinson's Disease mouse model. The results showed that Amomum tsaoko extract improved motor and constipation symptoms, reduced dopaminergic neuron loss, and lowered inflammation-related gene expression. It also enhanced gut barrier function and reversed gut dysbiosis, increasing beneficial bacteria while decreasing harmful ones. The study is very comprehensive and uses a holistic approach. The results are robust, well-supported by multiple types of evidence, and demonstrate a significant potential for Amomum tsaoko extract as a therapeutic intervention for PD. Using diverse behavioural, histological, and molecular analyses strengthens the conclusions.
The three main weaknesses are:
- The poor characterization of the extract. The authors mention the main compounds, found in doi.org/10.1016/j.fochx.2025.102177, where previous studies in Amomum tsaoko oil, seeds and fruit found a considerably higher number of compounds (please check https://doi.org/10.1007/s11101-021-09793-x, doi: 10.3390/foods11101402, DOI: 10.1039/c9ra07988b) Could the authors explain?
- While a holistic approach (i.e., using multiple potential active compounds) using an extract is pragmatic, allowing one to check for benefits, it does not allow one to understand what compounds are responsible for positive action. Could the authors expand the discussion to include previous works where single compounds from Amomum tsaoko were observed to be beneficial?
- While promising, the study could benefit from more mechanistic insights, including the molecular pathways involved in the observed effects. The authors could suggest possible mechanisms and possible future work to address this.
Please double-check for misspellings and grammar issues.
Author Response
Response to comments of Reviewer #2
Comments and Suggestions for Authors
The manuscript "Correlations between amelioration of rotenone-induced Park-inson's symptoms by Amomum tsaoko flavonoids and gut microbiota in mice" by Liu et al. reports the use of an Amomum tsaoko extract, assessed for composition in a previous publication (https://doi.org/10.1016/j.fochx.2025.102177), on rotenone-induced Parkinson's Disease mouse model. The results showed that Amomum tsaoko extract improved motor and constipation symptoms, reduced dopaminergic neuron loss, and lowered inflammation-related gene expression. It also enhanced gut barrier function and reversed gut dysbiosis, increasing beneficial bacteria while decreasing harmful ones. The study is very comprehensive and uses a holistic approach. The results are robust, well-supported by multiple types of evidence, and demonstrate a significant potential for Amomum tsaoko extract as a therapeutic intervention for PD. Using diverse behavioural, histological, and molecular analyses strengthens the conclusions.
The three main weaknesses are:
- The poor characterization of the extract. The authors mention the main compounds, found in doi.org/10.1016/j.fochx.2025.102177, where previous studies in Amomum tsaoko oil, seeds and fruit found a considerably higher number of compounds (please check https://doi.org/10.1007/s11101-021-09793-x, doi: 10.3390/foods11101402, DOI: 10.1039/c9ra07988b) Could the authors explain?
Response: Thank you very much for your comments and recognition of our work. I have reviewed the three articles you shared. It is true that the number of compounds found in the study of Amomum tsaoko oil, seeds and fruits mentioned in the articles as you said is much higher, but the types of compounds it focuses on are different from those in this study. For example, https://doi.org/10.1007/s11101-021-09793-x focus on the compounds isolated and identified from Amomum tsaoko; doi: 10.3390/foods11101402 focus on the main components of Amomum tsaoko essential oils; DOI: 10.1039/c9ra07988b focus on the seeds of Amomum tsaoko Crevost et Lemaire ethanol extract; However, this study mainly focuses on the study of ATFs, so the number of compounds involved is not as large as those mentioned in the three articles. The preparation method of ATFs has been briefly described in the “4. Materials and Methods” section of the article with reference to previous studies. According to your suggestion, we have made corresponding modifications to“4.2” , as follows.
The provenance of the AT fruits was Gongshan County of Yunnan Province. The ATFs were prepared in accordance with the published method from our group [73]. The AT was pulverized and then extracted using a sonicator. Subsequently, the upper sample solution was prepared at a specific concentration and pH. Adsorption and desorption were conducted with HPD300 macroporous resin, and the 20% and 30% ethanol eluates were concentrated and vacuum-dried, and stored. The 20% and 30% ethanol-eluting fractions contained more than 90% flavonoid compounds (based on the NaNO3-Al (NO3)3-NaOH colorimetric method) with a relative abundance of 73.83% (based on plant-wide targeted metabolomics). The components of the ATFs were determined using UPLC-QTRAP-MS/MS [73]. The ATFs obtained from the preparation primarily consisted of (+)-epicatechin, isoquercitrin, astragalin, kaempferol-3-O-rutinoside, procyanidin B2, vanillin, rutin, and other compounds (Table S1).
- While a holistic approach (i.e., using multiple potential active compounds) using an extract is pragmatic, allowing one to check for benefits, it does not allow one to understand what compounds are responsible for positive action. Could the authors expand the discussion to include previous works where single compounds from Amomum tsaoko were observed to be beneficial?
Response: We extremely agree with your comments. The focus of this study is to evaluate the ameliorative effect and mechanism of ATFs on Parkinson's disease as a whole. After literature research, there have been no reports of AT and single compounds extracted from AT used in Parkinson's disease research. However, some flavonoids in AT are also found in other plants or drugs, and the relationship between some single compounds and Parkinson's disease has been reported. The second paragraph of the discussion section of this manuscript discusses the research progress on the effects of epicatechin, isoquercitrin, astragalin, kaempferol-3-O-rutinoside, procyanidin B2, and rutin on PD. The specific content is as follows:
The ATFs employed in this work included epicatechin, isoquercitrin, astragalin, kaempferol-3-O-rutinoside, procyanidin B2, and rutin. All of these flavonoid monomers have shown potential ameliorative effects on PD in previous studies. For instance, epicatechin-rich green tea polyphenols have been shown to offer some degree of protection to dopaminergic neurons in rat models of PD [25]. Isoquercitrin has been shown to exert a palliative effect on PD by enhancing the anti-apoptotic capacity of human proximal tubule epithelial cells (HK2 cells) [26]. Astragalin has been shown to counteract lipopolysaccharide-induced neuroinflammation in mice by reducing nitric oxide synthesis and pro-inflammatory cytokine production [27]. Both kaempferol and its derivatives have been shown to directly reduce the aggregation of α-syn and the production of Lewy bodies and to suppress central nervous system inflammation and oxidative stress [28]. Proanthocyanidin-rich extracts have been shown to activate cellular antioxidant mechanisms and alleviate mitochondrial dysfunction, consequently slowing down dopaminergic neuronal death induced by related toxins, which in turn mitigates PD symptoms [29]. Rutin has been demonstrated to exert neuroprotective effects through modulating mitochondrial autophagy by attenuating oxidative damage and depolarizing the mitochondrial membrane potential [30]. These previous studies corroborate the results of the present study. There are many kinds of compounds in ATFs, and the interactions between them are also complex. Therefore, it is of great significance to evaluate the overall effect of Amomum tsaoko total flavonoids on PD.
- While promising, the study could benefit from more mechanistic insights, including the molecular pathways involved in the observed effects. The authors could suggest possible mechanisms and possible future work to address this.
Response: Thank you for your professional review and we fully agree with you. The focus of this study is to evaluate whether ATFs can improve Parkinson's disease and the link between this improvement effect and gut microbiota. The topic of the paper focuses on the correlation between "symptoms and microbes. In terms of future research plans, in addition to further exploring the anti-Parkinson mechanism of ATFs from classical signaling pathways (such as PI3K/Akt, MAPK, etc.), we also plan to further explore the components that play a major role in ATFs to lay a stronger foundation for future in-depth (clinical) research. According to your suggestion, we have added the relevant contents of the future research plan in the conclusion part. The revised conclusion is now presented as follows, please review it.
This study presents the novel finding that ATFs exert a protective effect against low-dose rotenone-induced PD in mice, including the amelioration of motor deficits and constipation symptoms, the preservation of DA neurons, a reduction in inflammatory responses, and the enhancement of the gut barrier. Notably, ATFs significantly influenced the structure and composition of the gut microbiota in PD mice. The correlation analysis between the gut microbiota and the PD phenotypic indicators provides important evidence for elucidating the possible mediating role of the gut microbiota in the amelioration of PD by ATFs. Several conditionally pathogenic and probiotic taxa may play an important role in the alleviation of PD by ATFs. In conclusion, the oral administration of ATFs may have potential as a modality for the prevention and treatment of PD. The anti-PD mechanism of ATFs may be related to the regulation of the gut microbiota; however, further in-depth investigations are required to determine the molecular mechanisms. In addition, the investigation and confirmation of the main active components of ATFs against PD are also of interest. The results of this study provide a theoretical basis for the future application of ATFs in clinical research, food products, and medicine.
Thanks again for your professional advice!
Following your suggestion, this manuscript has been polished to facilitate a clearer expression of the study content, as documented below:
References in this response:
- Guo, S.; Yan, J.; Yang, T.; Yang, X.; Bezard, E.; Zhao, B. Protective effects of green tea polyphenols in the 6-OHDA rat model of Parkinson's disease through inhibition of ROS-NO pathway. Biol. Psychiatry2007, 62, 1353-1362.
- Magalingam, K.B.; Radhakrishnan, A.; Ramdas, P.; Haleagrahara, N. Quercetin glycosides induced neuroprotection by changes in the gene expression in a cellular model of Parkinson's disease. J. Mol. Neurosci.2015, 55, 609-617.
- Kim, E.H.; Shim, Y.Y.; Lee, H.I.; Lee, S.; Reaney, M.J.T.; Chung, M.J. Astragalin and Isoquercitrin Isolated from Aster scaber Suppress LPS-Induced Neuroinflammatory Responses in Microglia and Mice. Foods2022, 11.
- Jin, S.; Zhang, L.; Wang, L. Kaempferol, a potential neuroprotective agent in neurodegenerative diseases: From chemistry to medicine. Biomed. Pharmacother.2023, 165, 115215.
- Tambe, M.A.; de Rus Jacquet, A.; Strathearn, K.E.; Hensel, J.A.; Colon, B.D.; Chandran, A.; Yousef, G.G.; Grace, M.H.; Ferruzzi, M.G.; Wu, Q. et al. Protective Effects of Polyphenol-Rich Extracts against Neurotoxicity Elicited by Paraquat or Rotenone in Cellular Models of Parkinson's Disease. Antioxidants2023, 12.
- Lai, X.; Zhang, Y.; Wu, J.; Shen, M.; Yin, S.; Yan, J. Rutin Attenuates Oxidative Stress Via PHB2-Mediated Mitophagy in MPP(+)-Induced SH-SY5Y Cells. Neurotox. Res.2023, 41, 242-255, doi:10.1007/s12640-023-00636-5.